# Molecular Mechanisms and Targeted Therapies of Advanced Basal Cell Carcinoma

**DOI:** 10.3390/ijms231911968

**Published:** 2022-10-08

**Authors:** Toshihiko Hoashi, Naoko Kanda, Hidehisa Saeki

**Affiliations:** 1Department of Dermatology, Nippon Medical School, Bunkyo-Ku, Tokyo 113-8602, Japan; 2Department of Dermatology, Nippon Medical School Chiba Hokusoh Hospital, Inzai 270-1694, Japan

**Keywords:** basal cell carcinoma, hedgehog signaling, p53, basal cell nevus syndrome, smoothened inhibitors, immune checkpoint inhibitors, clinical trials, biomarkers

## Abstract

Among human cutaneous malignancies, basal cell carcinoma is the most common. Solid advances in unveiling the molecular mechanisms of basal cell carcinoma have emerged in recent years. In Gorlin syndrome, which shows basal cell carcinoma predisposition, identification of the patched 1 gene (PTCH1) mutation was a dramatic breakthrough in understanding the carcinogenesis of basal cell carcinoma. PTCH1 plays a role in the hedgehog pathway, and dysregulations of this pathway are known to be crucial for the carcinogenesis of many types of cancers including sporadic as well as hereditary basal cell carcinoma. In this review, we summarize the clinical features, pathological features and hedgehog pathway as applied in basal cell carcinoma. Other crucial molecules, such as p53 and melanocortin-1 receptor are also discussed. Due to recent advances, therapeutic strategies based on the precise molecular mechanisms of basal cell carcinoma are emerging. Target therapies and biomarkers are also discussed.

## 1. Introduction

Basal cell carcinoma (BCC) is one of the most ordinary malignant tumors and its incidence continues to grow in many countries [1]. Moreover, BCCs are one of the most common cutaneous malignancies and account for approximately 80% to 90% of skin cancers in Caucasians [2,3]. Sunlight exposure is the most crucial risk factor, and radiation, age, and skin type [4] are also associated as possible causes. BCCs are defined as indolent growing tumors, which very rarely metastasize but can cause tissue destructions if they are not treated adequately [5]. Common metastasizing sites were regional lymph nodes (53%), lungs (33%), and bone (20%) according to a retrospective analysis of published metastatic BCC (mBCC) from 1981 to 2011, and the mean time between the occurrence of the primary tumor and signs of metastasis was about 9 years [6]. BCCs that show specific features can be classified as having a high metastatic potential. Concretely, tumors located in the midline of the face or ear, a tumor that has been present for a long period of duration, large tumor diameter size, and history of previous radiation exposure are such features [7,8]. Rare cases have reported BCC having intravascular invasion, showing higher metastatic rates than BCC cases without intravascular invasion. Systemic treatments are required for such mBCCs [9,10].

In this article, we review the clinical presentation and histologic features of BCCs. The molecular mechanisms that underpin BCCs and the systemic treatments based on the molecular mechanisms regulating the pathogenesis and development of BCCs are then described.

## 2. Clinical Presentations

Clinical manifestations may vary widely in BCCs, and BCC lesions usually grow slowly, are non-healing, and might show bleeding or ulceration [11,12,13,14].

### 2.1. Nodular BCC

The most frequent clinical subtype is nodular BCC [15]. The classical clinical description is that of a well-defined, pearly, shining, translucent and smooth nodule or papule with distinct, arborizing, and dilated blood vessels/telangiectasias often with ulcerations or erosions with rolled boundaries (Figure 1A). Nodular BCC may grow and crusting may occur over a depressed center. Hemorrhage with trivial trauma is frequent then the lesion sometimes may ulcerate, as known as rodent ulcer.

### 2.2. Superficial BCC

The second most frequent clinical subtype is superficial BCC, accounting for up to 15% of BCCs [15]. Clinical features appear as an annular, well-defined, thin plaque or patch. Superficial BCCs preferentially occur on the trunk and extremities, although other subtypes are usually seen on the head and neck [11,12,13,14,15].

### 2.3. Morpheic/Morpheaform/Sclerosing BCC

Morpheic BCC accounts for 5 to 10% of BCCs [15]. This type of BCCs are called morpheic or sclerosing because these show poorly defined lesions, scar-like features, infiltrative, or shining plaques, which might be flat or sometimes atrophic including telangiectasias, erosions, or small crusts (Figure 1B).

### 2.4. Pigmented BCC

Pigmented BCC appears as black nodules or papules sometimes with ulceration. Clinically, pigmented BCC can mimic melanoma and a histopathological examination is the only way to definitively diagnose BCC, to exclude the possibility of melanoma, and to decide on appropriate treatment [16]. Pigmented BCC is an unusual morphologic manifestation and has a low proportion of cases at 6.7% in Caucasians [17,18]. On the contrary, pigmented BCCs are the most frequent and typical clinical feature of Asian origin, especially in Japanese, and 88.3% of total BCCs in Japanese patients are pigmented [19].

## 3. Histopathology

The histopathology of BCC is generally distinguished by lobules of basaloid cells harboring large and hyperchromatic nuclei as well as scant cytoplasm, which are all accompanied by a fibromyxoid stroma and sometimes by tumor retraction spaces. The stroma surrounding BCC contains microvessel proliferations, linking to aggressive attitude [12,20].

Histopathologic subtypes of BCCs are classified according to the probability of the recurrence. BCCs with a low risk of recurrence are nodular, superficial, pigmented, infundibulocystic, and fibroepithelial, which show indolent attitudes [7]. In contrast, BCCs with a high risk of recurrence are morpheic/sclerosing, micronodular, infiltrating, basosquamous, and sarcomatous differentiated BCCs, which show aggressive histologic features [7]. Alternatively, BCCs were also suggested to be classified into ‘easy to treat’ and ‘difficult to treat’ [21,22,23]. We should bear in mind that clinical and histopathological classifications are distinct.

### 3.1. Nodular BCC

Islands of basaloid tumor cells or large nests with disorderly cell arrangements are observed in nodular BCCs [13]. Peripheral palisadings are usually seen. The tumor stroma with spindle cells is cleaving and is mucoid or myxoid, occasionally containing amyloid deposits (Figure 2A). The nests of tumor cells infiltrate deep into the dermis. Apoptosis also can be found in the center. According to the secondary findings such as keratotic, cystic/nodulocystic and adenoid BCCs, nodular BCC has several subtypes [12,20].

### 3.2. Superficial BCC

Superficial BCC appears as small lobules or islands containing basaloid tumor cells with peripheral palisading. Linking to the epidermis associated with a lichenoid or bandlike inflammatory cell infiltration within a myxoid stroma, the tumor localizes in the superficial dermis. Superficial BCCs can be multicentric and a mixed pattern tumor, with nodular, micronodular, or infiltrating subtype [12,13,20].

### 3.3. Morpheic/Sclerosing BCC

Morpheic/sclerosing BCC is characterized by very thin strands of tumor cells. These are buried in a collagenous stroma, however, tumor stroma cleftings are a rare phenomenon (Figure 2B). Morpheic/sclerosing BCC infiltrates deeply but differs from the infiltrating BCC by the stromal features. Infiltrating BCC lacks the collagenous stroma, which means sclerosing [12,20].

### 3.4. Micronodular BCC

Micronodular BCC is composed of dispersed micronodules of tumor cells. These tumor nests extend into the deeper dermis, and sometimes into the subcutaneous tissues. Satellite like arrangement of punctate nodules with irregular contours can be seen. Those are separated by normal dermal collagen and lined by a thin margin of stroma.

### 3.5. Infiltrating BCC

Infiltrating BCC is a subtype composed largely of chords of tumor cells which deeply infiltrate with angulated edges. This type of BCC shows a permeating or an irregular invasion pattern at the tumor periphery. Infiltrating BCC frequently overlaps with morpheic/sclerosing BCC and can be found with a nodular part [12,20].

### 3.6. Pigmented BCC

Pigmented BCC is a variant of superficial or nodular BCC, containing melanin pigment derived from activated melanocytes within the tumor nests, residing within the tumor cells or the melanophages [12,13,20,24]. In pigmented BCCs, tumor cell, as well as melanophages in the adjacent dermis, harbor pigment present within those cells in pigmented BCCs. This shows the reason for the black appearance of the tumor [25].

## 4. Molecular Mechanisms

Nevoid basal cell carcinoma syndrome (NBCCS), also known as Gorlin Goltz syndrome, is an autosomal dominant disorder that manifests in the pits of the palms and soles, jaw keratocysts, various other tumors, and developmental abnormalities as well as multiple BCCs [26]. The candidate gene was narrowed down to chromosome 9q22.3 [27,28,29]. Subsequently, loss of heterozygosity in the region was proved to be quite crucial for the pathogenesis of sporadic BCCs as well as of NBCCS [30]. These data show that the gene might be a tumor suppressor. Thus, unveiling the molecular mechanisms of BCCs clearly relies on a careful analysis of NBCCS. Here we discuss the molecular mechanisms of BCCs; however, sporadic BCCs have several histopathological classifications as described above. What regulates the histological subtypes? Biomarkers might be of help and will be discussed later.

### 4.1. The Sonic Hedgehog Pathway

Molecular analysis revealed that the gene corresponds to a human homolog of Drosophila patched [31,32]. Now named patched 1 PTCH1, the gene encodes a twelve-pass transporter like transmembrane protein and a receptor of the hedgehog ligand (Sonic hedgehog (SHH), Desert hedgehog (DHH) and Indian hedgehog (IHH)). In vertebrate development, the hedgehog pathway is crucial [33]. The SHH ligand binds to the PTCH1 receptor, inhibiting it and allowing signaling through the hedgehog pathway (Figure 3). Mice overexpressing SHH show features of NBCCS and develop multiple BCCs [34,35]. The dysregulation of PTCH1, in other words, if its inhibitory functions are gone, is known to be crucial in the carcinogenesis of BCC.

Somatic mutations in smoothened (SMO), a seven-transmembrane protein immediately downstream of PTCH1, were activated in sporadic BCCs. Transgenic mice overexpressing mutant SMO developed skin abnormalities similar to bona-fide BCCs [36]. PTCH1 inhibits SMO function when the hedgehog ligand is missing [37]. PTCH-mediated repression of SMO is relieved on ligand binding. SMO transduces the signal to a SUFU (suppressor of fused)-GLI (glioma associated oncogene) complex residing in the cytoplasm. Then, it activates GLI transcription factors [38]. Mutated PTCH1, found in NBCCS, cannot inhibit SMO, in other words, SMO is constitutively active.

SMO is thought to be a target of pharmacological therapeutic strategy. Because SMO serves as an oncogenic molecule and has drug-binding pockets in its transmembrane helices [39,40]. Gain-of-function mutations in the SMO gene (L412F, S533N, W535L, and R562Q) activate the GLI transcription network [36,39,41]. In the drug-binding cavity, R400, D473 and E518 are key amino acid residues. Cyclopamine, which is a naturally occurring alkaloid and is found in the corn lily, serves as an SMO antagonist [42]. The acquired D473H SMO mutation causes resistance to vismodegib and sonidegib, which are engineered molecules from cyclopamine [39]. SMO serves as a proto-oncogene according to these findings.

In hereditary and in sporadic BCCs, mutations in several genes in the hedgehog pathway have been identified. Mutations in SMO as well as PTCH1 have been described in a subset of patients with sporadic BCC [36,43,44]. These mutations mostly consist of C to T substitutions at a dipyrimidine site, belonging to the so-called ‘ultraviolet (UV) signature’ mutations. PTCH2 shares structural similarities with PTCH1 and has a minor compensatory role in the hedgehog pathway. PTCH2 mutations were found in some cases of sporadic BCCs [45]. PTCH2 is overexpressed in both hereditary and sporadic BCCs, indicating that PTCH2 is a direct gene target of hedgehog signaling, and that PTCH1 may negatively regulate PTCH2 [46]. Loss of PTCH2 has been reported to contribute to enhanced tumorigenesis in PTCH1 haploinsufficient mice [47]. By sequestering GLI transcription factors in the nucleus and in the cytoplasm, SUFU is the main repressor of the mammalian hedgehog signaling pathway [48]. As a negative regulator of the hedgehog pathway that acts by sequestering all three hedgehog homologs with similar affinity to that of PTCH1 protein, the transmembrane hedgehog interacting protein (HIP) was identified [49]. HIP was dysregulated in the nodular type BCCs [50]. There are three GLI proteins, which are GLI1, GLI2 and GLI3. These are activated by SMO, and then can activate or repress hedgehog pathway target genes [39]. Overexpression of GLI proteins in mice has been proven to induce BCCs [51,52,53]. Moreover, hedgehog signaling has been shown to be crucial for carcinogenesis of BCC because conditionally GLI2-expressing mice show BCC regression when GLI2 expression is inactivated [53]. KIF7, a member of the kinesin-4 family, was identified as a GLI-interacting protein [54]. KIF7 binds to GLIs and regulates their degradation and stability. Then KIF7 controls GLI-mediated transcription [55]. KIF7 has SUFU-independent and -dependent regulatory functions. While KIF7 promotes hedgehog pathway activity through the dissociation of SUFU-GLI2 complex, it also represses the hedgehog target genes in the absence of SUFU. Inactivation of either KIF7 or SUFU alone in the epidermis cannot promote BCC carcinogenesis, although their simultaneous deletion can induce BCC. These results demonstrate the distinct and overlapping roles of SUFU and KIF7 in GLI2 regulation during tumorigenesis [56]. These studies suggest that overactivation of hedgehog signaling is necessary and perhaps sufficient for the development of BCCs.

### 4.2. Other Molecules

#### 4.2.1. TP53

The *TP53* gene is the most commonly mutated tumor suppressor in malignancy, and has been described as ‘guardian of the genome’ [57,58]. The gene *TP53* encodes the protein p53; mutations of *TP53* are found in a wide variety of tumors, and abrogation of this pathway is found in many human malignancies [59]. p53 senses genotoxic injury and causes arrested cell division, allowing DNA repairment prior to replication. p53 also induces apoptosis in order to eliminate potentially malignant cells if extensive DNA damage occurs. Mutations of *TP53* are found in BCC tumors but not in the germ line, in 44% to 100% of the BCCs studied [60,61,62,63,64,65,66,67,68].

Li-Fraumeni syndrome harbors *TP53* germ line mutation and shows a cancer predisposition [69]. Accordingly, BCC is not predisposed to by this syndrome, indicating that *TP53* mutation is not necessary for BCC carcinogenesis. *TP53* mutation may be a secondary event occurring after BCC carcinogenesis. On the contrary, p53 loss enhances BCC carcinogenesis if the hedgehog pathway is already activated in mice [70]. Currently, this discrepancy has not been fixed. The cell cycle checkpoint gene 14-3-3σ, which is under the regulation of p53, is silenced in BCC [71,72]. The protein encoded by 14-3-3σ is essential for keratinocyte senescence. Thus, loss of 14-3-3σ may contribute to the growth of BCC [73]. Overexpression of p53 in BCCs has been reported to be associated with aggressiveness [67,74,75,76,77,78]. Of those, necrosis was significantly associated with overexpression of p53 in BCCs [78]. p53 expression may have some role in determining the aggressiveness of BCCs.

#### 4.2.2. Melanocortin 1 Receptor

Differences in the distribution and type of melanin produced in cutaneous melanocytes regulate wide variations of human hair and skin colors [79,80,81]. Melanin consists of two distinct types: one is the brown/black eumelanin, which is photoprotective. Another is the yellow/red pheomelanin, which is photosensitive. Individuals with red hair and light skin have a predominance of pheomelanin and/or a dysregulated ability to produce eumelanin. The red hair phenotype can be deeply associated with a single gene mutation which encodes the melanocortin 1 receptor (MC1R). Single gene mutations of MC1R are also responsible for freckling, and poor tanning response to UV as well as fair skin color.

MC1R, a membrane-anchored protein consisting of 317 amino acids, has 7 transmembrane spanning domains and is G-protein coupled receptor expressed on the surface of melanocytes [79,80]. In persons with a history of BCC, MC1R of red hair color mutations are overexpressed [82]. These mutants all bind α-melanocyte-stimulating hormone (α-MSH), which leads to production of melanin, but are unable to activate adenylate cyclase [83].

Several studies indicate the association of MC1R mutations with the risk of BCC, and show that a fair skin color in addition to MC1R mutations greatly increases this risk [82,84,85]. Melanocytes with loss-of-function MC1R mutations have an impaired ability to repair UV-induced DNA damage *in vitro*, however, their melanin content is not associated with this effect [86]. It is then possible that the same effect occurs in the carcinogenesis of BCCs. BCC risk in the two or more MC1R variants is reported to be increasing independently of skin type [81,85,87,88,89,90,91]. Interestingly, the frequency of P1315L mutation in BCCs is not associated with the MC1R genotype [89]. This indicates that the MC1R genotype is associated with the risk of BCC, independent of red hair characteristics.

## 5. Cell of Origin

The origin of BCC is still a controversial but intriguing matter due to conflicting evidences [92]. Several mouse models have been investigated so far to unveil the origin of BCC. Constitutively active SMO mutant in hair follicle bulge stem cells and in their transient amplifying progenies could not induce BCC. However, BCC arises from resident progenitor cells of the interfollicular epidermis and from the upper infundibulum when constitutively active SMO mutants are introduced [93]. Overexpression of GLI2 in stem cells of resting hair follicles induced nodular BCC. In contrast, overexpression of GLI2 in epidermis formed superficial BCCs [94]. BCC exclusively originated from keratin 15-expressing stem cells of the follicular bulge using fate tracking of X-ray induced BCCs in PTCH1^+/-^ mice [95]. Loss of p53 enhanced BCC carcinogenesis from the bulge and produced BCCs from the interfollicular epidermis by enhancing SMO expression, at least in part, thus, loss of p53 may be a primary event in BCC formation through SMO upregulation.

Using BerEP4 immunohistochemistry, BCC originates from the basal cell of epidermis, located at the interfollicular epidermis and infundibulum of the hair follicle located along the basal layer, suggesting the notions that BCC originates from the basal cell layer of the interfollicular epidermis, not from the outer root sheath or of the hair follicles. These data indicate that BCCs may arise from basal keratinocytes of the interfollicular epidermis or of the hair follicle [96].

## 6. Therapy

The goal of treatment is the complete removal of BCCs and to avoid dysfunctions as well as disfigurements. Surgical removal is the treatment of choice considered as the traditional mainstay in most cases [7,21,97,98,99]. BCCs occur mainly in exposed areas including face and scalp, and such locations often make surgery difficult. Mohs micrographic surgery is the gold standard for aggressive or ‘difficult to treat’ BCCs, especially in difficult anatomic sites. Radiation therapy can be an alternative option for certain patients. For indolent BCCs, topical therapy with 5-fuorouracil or imiquimod is also approved and shows amazing results [100,101,102,103,104]. Although photodynamic therapy (PDT) has not been approved by the FDA, PDT has shown to be more effective for superficial BCCs than for nodular BCCs [105,106]. However, a subset of advanced BCC cases requires systemic therapy [7,21,97,98,99].

### 6.1. Conventional Chemotherapy

Historically, conventional cytotoxic chemotherapy in metastatic or locally aggressive BCCs has been used [107,108,109,110]. However, data are limited, primarily based on case reports, and no drug was approved for metastatic or locally aggressive BCCs by the US Food and Drug Administration (FDA).

Systemic platinum-based chemotherapy achieved an overall response rate (ORR) of up to 77% to 83%, with a complete response in up to 45% of treated lesions of locally aggressive BCCs or mBCCs, although the duration was several months [111,112]. Other case reports with platinum-based chemotherapy showed a low response rate (RR) (20-30%) and a short duration of response [113]. These reports consistently suggest that a platinum-based regimen should be considered for the systemic chemotherapy of BCCs.

In 2015, the term ‘locally advanced BCC’ (laBCC) was first used and refers to a complex clinical scenario in which the tumor was untreated for a long time or the tumor has had repetitive treatment failures as well as recurrences. The same term is also used to describe the presence of an extensive tissue destruction by BCC in the surrounding area that makes it quite difficult to treat by surgery or radiotherapy; in other words, only systemic drug therapy is prospecting [98].

### 6.2. FDA Approved Hedgehog Inhibitors

As mentioned above, the main driver of BCC carcinogenesis and progression is the constitutive activation and dysregulation of the hedgehog pathway. The hedgehog pathway is involved in many situations of fetal development, and is also strictly regulated after birth [114]. Hedgehog pathway reactivation, caused by several associated mutations in factors of the hedgehog pathway, induces uncontrolled proliferation of the malignant cells leading to tumor formation. NBCCS patients have germline mutations in PTCH1 and show BCC growth at a very early age [31]. Therefore, BCCs in NBCCS patients are currently mostly treated with hedgehog inhibitors (HHIs). Moreover, HHIs have determined a paradigmatic shift toward laBCCs or mBCCs [115]. As an HHI, SMO antagonist is promising and the first SMO antagonist is a naturally occurring alkaloid called cyclopamine, found in the corn lily [42]. However, poor oral bioavailability, acid sensitivity and some degrees of specificity constrict the clinical usage of cyclopamine [116,117]. Recently, several SMO inhibitors, including vismodegib (GDC-0449) and sonidegib (Erismodegib, NVP-LDE-225, LDE-225), have been engineered from cyclopamine and have gained success as targeted clinical cancer therapy [118] (Table 1).

#### 6.2.1. Vismodegib

Vismodegib is the SMO antagonist approved by the FDA in 2012 based on a worldwide phase II study (ERIVANCE; NCT00833417) [119,120]. Vismodegib showed RR of 30% (laBCCs) and 43% (mBCCs) with a median response duration of 7.6 months [121]. The updated report showed RR of 48.5% (mBCC) and 60.3% (laBCC) at 39 month, and median response duration of 14.8 months (mBCC) and 26.2 months (laBCC) [119]. Most patients experienced adverse events (AEs) by vismodegib including muscle spasms, alopecia, taste loss, weight loss, decreased appetite, fatigue, nausea, or diarrhea [122,123]. BCC patients treated with vismodegib have an increased risk of developing cutaneous SCCs [124]. Squamous differentiation was observed in some BCC metastasis, and the activating SMO mutation c.1234C > T was found twice [124,125]. SMO mutations c.1234C > T provided resistance against vismodegib [126].

#### 6.2.2. Sonidegib

Sonidegib is an orally administered SMO inhibitor which is structurally distinct from vismodegib. It was approved by the FDA in 2015 for the treatment of laBCC or recurrent BCC [127]. Approval was based on the results of the phase II BOLT trial (NCT01327053) [128,129]. Note that for mBCCs, sonidegib has not been approved. Patients were randomized to receive either 200 mg or 800 mg of sonidegib daily. Among those with laBCCs, the objective RR after 30 months were 38% (800 mg) and 43% (200 mg). In the patients with mBCCs, the objective RR were 17% (800 mg) and 15% (200 mg) [129]. Objective RR in the 200 mg group were sustained at 56.1% (laBCC) and 7.7% (mBCC) at 30-month follow-up [130]. The 200 mg dose also exhibited a lower rate of grade 3/4 AEs (31% vs. 56%) and AEs leading to drug discontinuation (22% vs. 36%) or dose reduction/interruption (32% vs. 60%) [130]. Most commonly reported AEs for sonidegib and vismodegib in the ERIVANCE [122] and BOLT [128] studies were muscle spasms, alopecia, and dysgeusia. Overall, time to onset of AEs was slightly later and less frequent with sonidegib.

### 6.3. Drug Resistance

Currently, no randomized controlled trial has compared vismodegib with sonidegib. Vismodegib appears to be the treatment of selection for mBCC, because vismodegib has FDA approval for mBCC and may have superior efficacy to sonidegib in treating mBCC [131]. The ORR with vismodegib and sonidegib was 47.6% at 21-month follow-up and 60.6% with 18-month follow-up, respectively. Treatment with vismodegib is known to show primary or secondary resistance in almost 20% of patients [132].

Again, D473 in the sixth transmembrane domain of SMO is one of the key amino acid residues having drug-binding cavity, to which vismodegib or sonidegib binds. Vismodegib and sonidegib are resistant to acquired D473H SMO mutation [39]. Approximately 50% of laBCCs initially show vismodegib resistance, while 21% of initial responders develop resistance later and experience disease progression or recurrence in a mean of 54.4 weeks [133]. This may be explained by the enrollment of only more aggressive mBCC [119]. The potency of hedgehog pathway inhibition by itraconazole was shown to be further enhanced when used in combination with HHI cyclopamine [134]. Therefore, inhibiting the hedgehog pathway through different mechanisms with a combination of itraconazole and HHI may overcome HHI resistance. Two patients with advanced BCC treated with itraconazole and vismodegib demonstrated disease control with acceptable toxicity profile [135]. A subset of patients under HHI shows progress of the disease due to primary resistance by the activation of noncanonical hedgehog pathway or additional signaling pathways [136]. Secondary HHI resistance is also observed due to additional mutations in the SMO drug-binding cavity [129,137]. Surprisingly, HHI treatment showed upregulation of MHC-I and -II and increased CD8+ T cells, which might enable an immune-related tumor response [138]. Thus, the combination of HHI with an immunotherapeutic agent is expected for synergic effects.

### 6.4. FDA Approved Immune Checkpoint Inhibitors

Immune checkpoints, including programmed death (PD)-1 and cytotoxic T cell lymphocyte-associated protein (CTLA)-4 receptors, are expressed on activated T cells [139]. PD-1 binds two ligands, the programmed death-ligand 1 (PD-L1) and programmed death-ligand 2 (PD-L2) on tumor cells [140]. CTLA-4 binds B7 (CD80/86) on antigen-presenting cells, which inhibits the immune cell activation. This may serve as a mechanism for inhibiting tumor infiltrating T cells, and has become a crucial therapeutic target for restoring immunity. Patients with advanced malignancies received significant benefit from studies of checkpoint inhibition with anti-CTLA-4 (ipilimumab) and anti-PD-1 (nivolumab and pembrolizumab) monoclonal antibodies. In fact, melanoma, a disease that is more likely to metastasize and become life-threatening as compared to BCC, was the first cancer to demonstrate clinical benefit from cytokines and immune checkpoint blockade [141].

Similar to melanoma, BCCs are generally characterized by UV damage, which translates into a high tumor mutational burden (TMB). Treatment with anti-PD-1 antibodies has shown a dramatic response in high TMB tumors [142]. Neoantigens, the putative targets of immune cells that recognize and eradicate neoplastic cells, is deeply associated with high TMB [141]. PD-L1 expression and TMB have been demonstrated to correlate with response to checkpoint inhibition [143]. BCCs have higher TMB with 47.3 median mutations/Mb than melanomas, which have TMB with 13.5 median mutations/Mb.

BCC was defined as a ‘cold tumor’, meaning that few immune cells can infiltrate tumor cells in BCC [14]. However, PD-L1 expression by tumor cells in BCCs ranges from 22% to 89.9%, while expression by tumor-infiltrating lymphocytes ranges from 82.0% to 94.9% [144,145,146]. Moreover, BCCs treated with checkpoint inhibition demonstrated greater PD-L1 expression in tumors (32% vs. 7%) and in TILs (47% vs. 18%), suggesting that treatment may induce PD-L1 expression, and that previously treated BCCs could possibly be more responsive to checkpoint inhibition [146].

Several case reports have been published of patients with mBCC or laBCC being treated with immune therapy, using ipilimumab, nivolumab or pembrolizumab, demonstrating meaningful and durable responses [147,148,149,150,151]. Thus, immune checkpoint inhibitors are promising toward laBCC and mBCC.

#### Cemiplimab

The first clinical trial using checkpoint inhibitors for BCCs was performed. That trial was a non-randomized and open-label study of pembrolizumab with or without vismodegib for advanced BCCs (NCT02690948) [146]. The ORR at 18 weeks was 44 % (pembrolizumab monotherapy group) and 29% (pembrolizumab with vismodegib group). Pembolizumab was active against BCCs, but the dual therapy group did not show significant superiority compared to the pembolizumab monotherapy group. Cemiplimab (REGN2810) is a fully human hinge-stabilized IgG4 high-affinity anti-PD-1 antibody that potently blocks functional interaction between PD-1 and PD-L1 [152]. The pharmacological difference between cemiplimab and nivolumab was unclear [153]. Comparisons of pembrolizumab and cemiplimab have not been published so far. Cemiplimab was developed for treating metastatic and locally advanced cutaneous SCCs [154], which is candidate for neither radical surgery nor radiation. SCCs and BCCs share some common features, including the high UV-induced TMB, indicating a strong rationale for response to immunotherapy [155]. The first pivotal study performed on cemiplimab monotherapy for laBCC and mBCC patients escaping on HHIs was a worldwide, open-label, phase II, and single-arm trial (Study 1620; NCT03132636) [156,157,158]. ORR in the laBCC cohort was 31%, including 6% CR and 25% PR, at a median follow-up of 15.1 months. Moreover, ORR in the mBCC cohort was 21.4% at a median follow-up of 38.9 weeks. Pretreatment and posttreatment biopsies were assessed for the expression of PD-L1 and MHC-I as well as TMB. Unfortunately, these analyses were not associated with clinical efficacy. Based on these results, cemiplimab gained FDA and European Medicines Agency approval as a second line treatment option for BCC patients who are resistant, progressive or are intolerant to HHI first line treatment.

Currently, HHI therapy is a first-line option for laBCC or mBCC [7]. Cemiplimab is indicated and recommended as a first-line after HHIs’ treatments, either when the HHI therapy becomes intolerable, or when the BCC progresses despite HHI treatment, or when the BCC is stable but does not improve after nine consecutive months of HHI therapy [159].

## 7. Biomarkers

BCCs are classified into several subtypes in terms of the histopathological features described above [7,12,20]. Moreover, BCCs are also classified into indolent and aggressive, or ‘easy to treat’ and ‘difficult to treat’ [21,22,23]. This classification is crucial for the treatment of BCCs, and molecular basis biomarkers are helpful [7,12,20,160].

Paraoxonase-2 (PON2) is an intracellular enzyme having antioxidant roles in reducing intracellular oxidative stress [161,162]. PON2 enzyme expression was significantly increased in infiltrating BCCs compared with nodular BCCs [163]. Nicotinamide N-methyltransferase (NNMT) is a human cytosolic enzyme catalyzing the N-methylation of nicotinamide, pyrimidines and other structural analogs, and plays a crucial role in biotransformation [164,165,166]. NNMT was reported to be overexpressed in BCCs compared with intact margins, although was expressed relatively lower in infiltrating BCCs compared with nodular BCCs [167]. EZH2 is a histone methyltransferase of the polycomb repressive complex 2) and is associated with hedgehog pathway [168]. EZH2 expression was reported to correlate with Ki67 and with aggressive BCC subtypes, including morpheaform, infiltrative and micronodular [169]. miRNA are small single-stranded non-coding RNAs regulating posttranscriptional modification of related proteins, and are associated with the development of neoplasms [160,170]. Serum expression levels of miR-34a in BCC patients was significantly lower than in healthy controls and prognosis was poor when expression levels of miR-34a was low [170].

Again, PD-1 binds two ligands, the PD-L1 and PD-L2 on tumor cells [140]. PD-L1 expression is expected to be a biomarker for PD-1 inhibition therapy and was reported to be a positive predictor of pembrolizumab for cutaneous squamous cell carcinomas [171,172]. However, as for BCCs, PD-L1 expression was not associated with the response to cemiplimab [146,156,171].

## 8. Clinical Trials

### 8.1. XmAb20717 Monoclonal Bispecific PD-1 and CTLA-4 Antibody

Immune checkpoint inhibitor combination therapies have been quickly adopted to overcome resistance and have been shown to achieve greater ORR due to different modalities of action of CTLA-4 and PD-1 [173]. However, this success is also accompanied by a higher incidence of immune-related AEs [173]. Bispecific antibodies comprise at least two distinct antigen-binding sites and therefore have two different specificities, binding two different antigens or two different epitopes on the same antigen [174]. Targeted combination therapy with a bispecific antibody is an advantageous strategy for overcoming systemic responses by pairing tumor-associated antigens in the context of T cell engagement [175]. XmAb20717 is a humanized bispecific monoclonal antibody for PD-1 and CTLA-4. Its safety and tolerability profile are confirmed by a phase I open-label study (NCT03517488). Its anti-tumor activity is currently recruiting, in which BCCs are included (Table 2).

### 8.2. Anti LAG-3 Antibody

Lymphocyte activation gene 3 (LAG-3), also known as CD223, is a single-pass transmembrane glycoprotein and is an immune checkpoint receptor associated with tumor escape and decreased T cell effector function [176]. LAG-3 is expressed widely on different cells: T cell subpopulations including activated CD4+ helper T cells and cytotoxic CD8+ T cells [177,178]. T cell activation is a crucial condition for LAG-3 expression on T cell subpopulations [179]. MHC-II interacts with LAG-3, inhibiting T cell receptor (TCR) binding, thus hindering T cell activation and promoting an anergic T cell state [180]. It also regulates T cells differentiation toward regulatory T cells and favors regulatory T cell immune suppressive functions [181]. LAG-3 inhibition can induce reversal of the anergy and restore T cell antitumoral functions. In BCC patients, the soluble cell-free variants of CTLA-4, LAG-3, PD-1/PD-L1 and T-cell immunoglobulin and mucin domain-3 (TIM-3), which were prominent co-inhibitory immune checkpoints, were measured. Those of all five molecules were significantly upregulated [182]. These results show that LAG-3 could be a biomarker of immunotherapy as well as a candidate for immune checkpoint inhibition.

### 8.3. Combination of HHIs with Immune Checkpoint Inhibitors

Recent studies have shown that HHI-induced BCC regression leads to drastic changes in the tumor microenvironment, recruiting cytotoxic T cells, activating adaptive immunity, and stimulating dynamic changes in the cytokine/chemokine network [138]. These results suggested a synergic effect between HHI and checkpoint inhibition in advanced BCCs. Results from a phase III study for advanced melanomas assessing the clinical efficacy of the combination of relatlimab (anti LAG-3 antibody) plus nivolumab versus nivolumab monotherapy have been published recently [183]. Median progression-free survival was significantly longer in the combination treatment arm (10.1 months) versus nivolumab monotherapy (4.6 months), indicating that dual immune checkpoint inhibition is preferable. Similarly, for patients with advanced BCC, an open-label, phase II trial is investigating cemiplimab in combination with pulsed sonidegib (NCT04679480) and nivolumab alone or in combination with relatlimab or ipilimumab (NCT03521830).

### 8.4. Itraconazole

Vismodegib and sonidegib are derived from a naturally occurring alkaloid called cyclopamine. The activation of the hedgehog pathway is inhibited by binding to SMO receptor in the drug-binding pocket. Inevitable secondary drug resistance to vismodegib and sonidegib have been reported and were attributed to several mutations occurring in the drug-binding pocket of SMO [184]. Itraconazole is an FDA-approved, well-known and commonly used agent for the treatment of fungal infections. Itraconazole also inhibits SMO accumulation in the cilium by binding to a site on the SMO receptor different to that of cyclopamine, and efficiently shows SMO agonist effects [134]. Itraconazole inhibits 14-α-lanosterol demethylase, an enzyme involved in the cholesterol biosynthesis in mammals [134]. For SMO activity and hedgehog pathway function, cellular cholesterol is essential [185]. However, itraconazole-induced hedgehog pathway inhibition seems to be independent from cholesterol synthesis [134]. It has been shown that itraconazole does not bind SMO like vismodegib or sonidegib. The inhibitory effect on the hedgehog pathway by itraconazole is attributed to the downregulation of SMO in the cilium. Itraconazole efficiently inhibits medulloblastoma and BCC growth. Moreover, itraconazole shows synergistic inhibition effects if used with cyclopamine [134]. The combination of itraconazole and arsenic trioxide efficiently inhibits the proliferation of hedgehog-driven medulloblastoma in the context of GLI2 overexpression or in the case of vismodegib resistance due to SMO mutations. However, itraconazole could not inhibit entire SMO mutations associated with vismodegib-resistance [186]. Currently, itraconazole is under phase II clinical trials for the treatment of BCC [187]. Itraconazole reduced cell proliferation by 45%, hedgehog pathway activity by 65%, and reduced tumor volume by 24% [187].

### 8.5. Other Inhibitors of the Hedgehog Pathway

#### 8.5.1. SMO Inhibitors

New drugs inhibiting the hedgehog pathway are under evaluation. Toxicity due to HHI is still a problem. Several SMO inhibitors other than vismodegib or sonidegib are currently being investigated in clinical trials.

#### 8.5.2. Taladegib

Taladegib (LY2940680), a phthalazine-based SMO inhibitor, is one of the few successful examples that can inhibit both the wild-type and the D473H mutant SMO in clinical trials [188]. Taladegib binds to a thin pocket of SMO. It shows weak interaction with D473, and D473H mutation does not interfere with its binding to SMO, which contributes to its inhibitory activity towards SMO-D473H mutant [189]. In a phase I, multicenter, open-label study, 47 patients with laBCC or mBCC were treated with taladegib. Note that sonidegib was not efficacious in an investigational study of advanced BCC patients previously treated with vismodegib [190]. ORR was 46.8%. Responses were observed in patients previously treated with HHI therapy (35.4%) and in HHI therapy-naive (68.7%) patients [191].

## 9. Conclusions

We reviewed the studies regarding the clinical presentation and histologic features of BCCs. We then described the molecular mechanisms that underpin BCCs and the systemic treatments based on the molecular mechanisms underlying the pathogenesis and development BCCs. The discovery of hedgehog signaling in BCCs is quite striking and targeted therapies have rapidly emerged. However, differences between nodular BCCs and morpheic BCCs, which are quite distinct, have not yet been unveiled. Drug resistances are also a big obstacle. Future investigations are eagerly awaited.

## Figures and Tables

**Figure 1 ijms-23-11968-f001:**
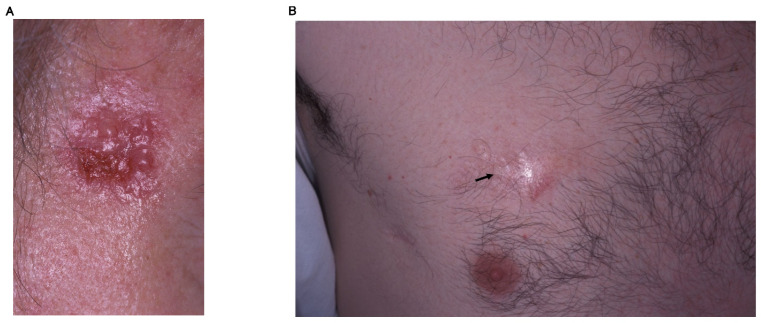
Clinical features of basal cell carcinoma. (**A**) Nodular BCC. Translucent small papulonodule accompanied by telangiectasias and ulceration in the center of the tumor. (**B**) Morpheic/sclerosing BCC. Scar-like shiny area is observed (arrow). Boundary is very inconspicuous.

**Figure 2 ijms-23-11968-f002:**
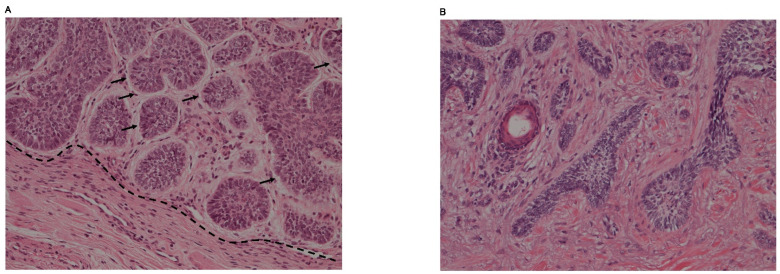
Histopathological features of basal cell carcinoma. (**A**) Nodular BCC. Basaloid tumor cells arranged in a palisading manner. Tumor stroma cleftings are also found (arrow). Boundary is relatively clear (dotted line, original magnification ×200). (**B**) Morpheic/sclerosing BCC. Chords or thin nests of basaloid tumor cells infiltrate into the dermis. Tumor stroma clefting is not clear. Boundary is also unclear (x200).

**Figure 3 ijms-23-11968-f003:**
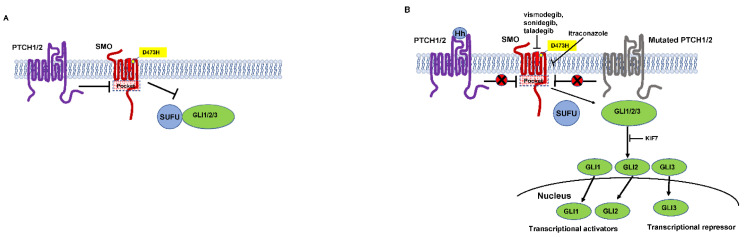
Hedgehog pathway basal cell carcinoma. (**A**) Inactive state. PTCH1/2 inhibits SMO in the absence of Hh. (**B**) Active state. On ligand (Hh) binding, PTCH1/2-mediated repression of SMO is relieved (left). Alternatively, SMO is constitutively relieved if PTCH1/2 is mutated (right). SMO inhibitors (vismodegib, sonidegib, and taladegib) bind to drug-binding pockets of SMO then inhibit SMO. However, D473H SMO mutation causes resistance to vismodegib and sonidegib. PTCH1/2, patched 1/2; Hh, hedgehog ligand; SMO, smoothened; pocket, drug-binding pocket; SUFU, suppressor of fused; GLI, glioma-associated oncogene.

**Table 1 ijms-23-11968-t001:** FDA approved target therapies.

Drug	Indication	Mechanism	ORR	Median Follow-Up Time	Identifier	AE
Vismodegib	laBCC	SMO inhibitor	60.3%	39 months	ERIVANCE; NCT00833417	muscle spasms, alopecia, taste loss, weight loss, decreased appetite, fatigue, nausea, diarrhea
	mBCC		48.5%	39 months		
Sonidegib	laBCC	SMO inhibitor	56.1%	30 months	BOLT trial; NCT01327053	muscle spasms, alopecia, dysgeusia
	mBCC		7.7%	30 months		
Cemiplimab	laBCC (HH failure)	anti-PD-1	31%	15.1 months	Study 1620; NCT03132636	fatigue, diarrhea, pruritus
	mBCC (HH failure)		21%	38.9 weeks		

Abbreviations: ORR, overall response rate; AE, adverse effect; BCC, basal cell carcinoma; laBCC, locally advanced BCC; mBCC, metastatic BCC; SMO, smoothened; PD-1, program death-1; HH, hedgehog. Note: Reported ORR in this table are from FDA-approved labels for each agent (Drugs@FDA).

**Table 2 ijms-23-11968-t002:** Clinical trials of target therapies.

Drug	Indication	Mechanism	Other Drugs	Mechanism	Phase	ORR	Identifier
XmAb20717	BCC	bispecific (anti-PD-1 and anti-CTLA-4)			Phase I		NCT03517488
Pemrolizumab	advanced BCC	anti-PD-1	vismodegib	SMO inhibitor	Phase I/II	44%	NCT02690948
						Pembrolizumab	
						29%	
						Pembrolizumab + vismodegib	
Cemiplimab	advanced BCC	anti-PD-1	sonidegib	SMO inhibitor	Phase II		NCT04679480
Nivolumab	laBCC/mBCC	anti-PD-1	relatlimab	anti-LAG-3	Phase II		NCT03521830
			Ipilimumab	anti-CTLA-4			
Itraconazole	BCC	SMO inhibitor			Phase II		NCT01108094

Abbreviations: ORR, overall response rate; BCC, basal cell carcinoma; laBCC, locally advanced BCC; mBCC, metastatic BCC; SMO, smoothened; PD-1, program death-1; CTLA-4, cytotoxic T-lymphocyte antigen-4; LAG-3, lymphocyte activation gene 3. Note: Reported ORR in this table are from FDA-approved labels for each agent (Drugs@FDA).

## Data Availability

Not applicable.

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
