# Peer review of "Molecular Mechanisms and Targeted Therapies of Advanced Basal Cell Carcinoma"

_ijms, 2022, doi:10.3390/ijms231911968_

Round 1

Reviewer 1 Report

The topic of this review is unique and the descriptions sound potentially interesting. Each section of this paper is reasonable and easy to understand.

if available, clinical photos of nodular BCC and morpheic/sclerosing BCC may be informative.

Author Response

Reviewer 1 Comments and Suggestions for Authors

The topic of this review is unique and the descriptions sound potentially interesting. Each section of this paper is reasonable and easy to understand. if available, clinical photos of nodular BCC and morpheic/sclerosing BCC may be informative.

Response: We thank the reviewer for the comment. We added clinical figures of nodular BCC and morpheic/sclerosing BCC.

Reviewer 2 Report

IJMS – 1868-485 Molecular mechanisms and targeted therapies of basal cell carcinoma.

This submission is a review of primary characteristics and recent additions to the mechanistic properties of basal cell carcinoma. A review of the English language would be helpful.

Need a brief connecting ‘introduction’ to the molecular mechanisms discussion. The manuscript currently jumps from describing the typical pathology of six basal cell tumor types to describing the mechanisms of nevoid basal cell carcinoma syndrome – which may or may not be of primary relevance for most BCC types.

The section currently labeled as “V” Cell of origin, should likely come prior to the current “IV” Molecular mechanisms”. The Molecular mechanism should be re-ordered to discuss the MC1R mutations, TP53, and then introduce and discuss what is currently known about the molecular mechanisms among tumors in those that are unfortunate to have nevoid basal cell carcinoma syndrome (NBCCS).

Section ‘VI Therapy’ appears to be discussing therapy only for NBCCS, which is fortunately a fairly rare disease.  The most common treatment for BCCs are removal, freezing/photo-dynamic therapy, topical applications (Imiquimod or 5-FU), or radiation.  Publications indicate that 95 to 99% of all BCCs are successfully treated solely with removal. Chemotherapeutic approaches only come into play in advanced scenarios including NBCCS.   

Along with re-ordering some of the content, changing the title of the review to “Molecular mechanisms and targeted therapies of advanced basal cell carcinoma” would better clarify the focus of the content. With some minor revisions this review will contribute to the understanding of advanced basal cell carcinoma.

Author Response

Reviewer 2 Comments and Suggestions for Authors

This submission is a review of primary characteristics and recent additions to the mechanistic properties of basal cell carcinoma. A review of the English language would be helpful.

Response: We thank the reviewer for the comment. Grammatical fallacies have been checked. Because of tight time frame, we simultaneously have a native speaker to review.

Need a brief connecting ‘introduction’ to the molecular mechanisms discussion. The manuscript currently jumps from describing the typical pathology of six basal cell tumor types to describing the mechanisms of nevoid basal cell carcinoma syndrome ? which may or may not be of primary relevance for most BCC types.

Response: We thank the reviewer for the comment. We adder a paragraph between “IV. Molecular mechanisms” and “1. The sonic hedgehog pathway” to discuss that gap.

The section currently labeled as “V” Cell of origin, should likely come prior to the current “IV” Molecular mechanisms”. The Molecular mechanism should be re-ordered to discuss the MC1R mutations, TP53, and then introduce and discuss what is currently known about the molecular mechanisms among tumors in those that are unfortunate to have nevoid basal cell carcinoma syndrome (NBCCS).

Response: We thank the reviewer for the comment. We totally agree with the reviewer. Initially, we planned to describe “Molecular mechanisms” AFTER “Cell origin”. However, in “Cell origin” paragraph, we must discuss about PTCH, SMO, GLIs. Therefore, we described “Molecular mechanisms” AFTER “Cell origin”. We are very sorry, but we would appreciate it if you could understand that situation. We discussed recent papers about MC1R and TP53.

Section ‘VI Therapy’ appears to be discussing therapy only for NBCCS, which is fortunately a fairly rare disease.  The most common treatment for BCCs are removal, freezing/photo-dynamic therapy, topical applications (Imiquimod or 5-FU), or radiation.  Publications indicate that 95 to 99% of all BCCs are successfully treated solely with removal. Chemotherapeutic approaches only come into play in advanced scenarios including NBCCS.  

Response: We thank the reviewer for the comment. We totally agree with the reviewer. We adder a paragraph between “VI. Therapy” and “I. Conventional chemotherapy” to discuss about surgical excision, including Mohs micrographic surgery, and topical treatment.

Along with re-ordering some of the content, changing the title of the review to “Molecular mechanisms and targeted therapies of advanced basal cell carcinoma” would better clarify the focus of the content. With some minor revisions this review will contribute to the understanding of advanced basal cell carcinoma.

Response: We thank the reviewer for the comment. We totally agree with the reviewer. Indeed, this paper is focused on the molecular mechanisms and therapies based on the molecular mechanisms, new title, “Molecular mechanisms and targeted therapies of advanced basal cell carcinoma” is quite appropriate.

Reviewer 3 Report

The manuscript “Molecular mechanisms and targeted therapies of basal cell carcinoma” is a review article regarding the clinical features, pathological features and treatments for basal cell carcinoma. The manuscript is generally well written, although sometimes there is an excessive use of too short sentences, few typos are present. The manuscript may be considered for publication upon addressing the following important concerns:

1.       In the introduction section there is an excessive use of too short sentences which makes the reading unpleasant. Please rewrite this part.

2.       There is something wrong with tables as they appear unclean. Please carefully fix them.

3.       Following the paragraph “clinical trials”, there is a lack of a paragraph which summarizes the recent research works which attempted to identify novel biomarkers or therapeutic targets for BCC (e.g.  PMID: 33210737; PMID: 34638427; PMID: 35743435). This is a crucial point since would help researches to understand where the actual studies are focused on.

Author Response

Reviewer 3 Comments and Suggestions for Authors

The manuscript “Molecular mechanisms and targeted therapies of basal cell carcinoma” is a review article regarding the clinical features, pathological features and treatments for basal cell carcinoma. The manuscript is generally well written, although sometimes there is an excessive use of too short sentences, few typos are present. The manuscript may be considered for publication upon addressing the following important concerns:

  1. In the introduction section there is an excessive use of too short sentences which makes the reading unpleasant. Please rewrite this part.

Response: We thank the reviewer for the comment. In the introduction section as well as in other parts, unpleasant short sentences have been avoided.

  1. There is something wrong with tables as they appear unclean. Please carefully fix them.

Response: We thank the reviewer for the comment. Character encoding/decoding errors in the tables have been fixed.

  1. Following the paragraph “clinical trials”, there is a lack of a paragraph which summarizes the recent research works which attempted to identify novel biomarkers or therapeutic targets for BCC (e.g. PMID: 33210737; PMID: 34638427; PMID: 35743435). This is a crucial point since would help researches to understand where the actual studies are focused on.

Response: We thank the reviewer for the comment. We cited three papers and added the new paragraph, “Biomarkers”. Then, currently being investigated biomarkers of BCCs were discussed.

Round 2

Reviewer 3 Report

The manuscript is improved and can be accepted for publication.

Author Response

We thank the reviewer for the comment.